



**The Loobos ecosystem first tower dataset: meteorology, turbulent fluxes and net ecosystem exchange (1996 to 2021)**

Hong Zhao[1], Han Dolman[2], Jan Elbers[a], Wilma Jans[3], Bart Kruijt[3], Eddy Moors[4], Henk Snellen[1], Jordi Vila-Guerau de Arellano[1], Wouter Peters[1, 5], Maarten C. Krol[1], Ronald Hutjes[3], Michiel van der Molen[1,*]

1: Environmental Sciences Group, Meteorology and Air Quality, Wageningen University, Wageningen, the Netherlands

2: Royal NIOZ, the Netherlands Institute for Sea Research, Den Burg, the Netherlands

3: Earth Systems and Global Change group, Wageningen University, Wageningen, the Netherlands

4: IHE Delft, Institute for Water Education, Delft, the Netherlands

5: Energy and Sustainability Research Institute Groningen (ESRIG), Centre for Isotope Research (CIO), University of Groningen, the Netherlands

a: Formerly at: Earth Systems and Global Change group, Wageningen University, Wageningen, the Netherlands

*: corresponding author: Michiel K. van der Molen (Michiel.vanderMolen@wur.nl)

**Abstract**

We describe a 25 years (1996-2021) observational dataset of meteorology, turbulent fluxes and net ecosystem exchange collected from the first tower at the Loobos site, the Netherlands (NL). This is one of the 17 first FLUXNET sites globally. The presented dataset contains six data streams, namely (1) the NL-Loo_BM stream including meteorological data: four-component radiation (radiation balance), air

temperature and relative humidity, wind information, precipitation and throughfall, photosynthetic active radiation, bole temperature and soil heat flux), (2) the NL-Loo_Profile stream containing vertical profiles of $CO_2$ mole fraction, $H_2O$ pressure, air temperature and relative humidity, (3) the NL-Loo_ST stream derived from the aforementioned two streams including total stored heat flux, $H_2O$ and $CO_2$ fluxes below the canopy, (4) the NL-Loo_EC stream including EC measurements of $CO_2$ flux, sensible heat and latent

heat fluxes, (5) the NL-Loo_Soil stream including vertical profiles of soil moisture and temperature and ground water level data, and (6) ancillary data including soil respiration, vegetation properties (i.e., tree height, stem width and dry aboveground biomass, Leaf Area Index, sap flow, needle foliage properties and the associated nutrient analysis) and ground water level. The data quality of these data streams is assured through standard operating procedures. To show the utility of gathering long-term and





comprehensive measurements, we present analyses of mean diurnal storage $CO_2$ flux, the trend of NEE over the last 25 years and the energy balance closure. Being one of the longest datasets of its kind in a temperate forest, this valuable dataset is anticipated to be used for investigating the performance of various gap-filling algorithms, semi-climatological trends including extreme climatic events (such as the heatwave of 2003 and the drought of 2018) and the role of forest ecosystem in the carbon, water and

energy cycle. Meanwhile, it is expected to be employed for validating modelled land-atmosphere $CO_2$ and turbulent exchange fluxes, verifying model assumptions and serving as ground truth for satellite data retrievals. The dataset is accessible at https://doi.org/10.5281/zenodo.15721310 under a CC-BY4 open use license, where it is published as an associated station-like site and the same data will also be available at the European Fluxes Database Cluster. Hence, the data will be committed to the FLUXNET Data

System Initiative too. It is noted that in 2021 a second tower was erected next to the first tower, which was labelled as an ICOS Ecosystem Class 2 site in 2023 (Van Der Molen et al., 2025). Here we describe the first tower's instrumentation and data processing up to a Level 1 product (derived variables and quality checks, but not gap-filled).

## 1    Introduction

In 1995, a tower was built in the Loobos forest area in the Netherlands to measure water, heat and momentum fluxes for investigating forest evapotranspiration using the eddy covariance method (referred to as EC from here on) (Dolman et al., 1998). Following the Kyoto negotiations, which sought to operationalize the United Nations Framework Convention on Climate Change by committing industrialized countries and economies in transition to limit and reduce greenhouse gas emissions

(https://unfccc.int/kyoto_protocol, last access: 15 December 2024). One of the key questions raised by the Kyoto Protocol was how to calculate the changes in carbon stocks associated with land use changes and forestry activity (Igbp Terrestrial Carbon Working Group, 1998). This required to also observe the carbon dioxide ($CO_2$) balance for forest ecosystems (Valentini, 2000). Consequently, since 1996, $CO_2$ flux measurements have been conducted in Loobos, which subsequently became one of the 17 first FLUXNET

sites globally (https://fluxnet.org/data/la-thuile-dataset/lathuile-data-summary/, last access: 15 December 2024).

The Loobos site is located near Kootwijk (52°9′59.50″N - 5°44′36.99″E). A 22 m tall tower was built on a small dune (please refer to photos in Appendix 0). The tower base is at 26.4 m above mean sea level. The main tree species is Scots pine (*Pinus sylvestris*). In all directions the forest extends for more than 1.5 km.

The forest was planted around 1911 on bare sand (Kadaster, 2025) to control the drifting of the sand and provide wood for the mining industry. Before planting, sand dunes had formed with heights between 2 and 10 m relative to the valleys in between (see Fig. 2 in (Van Der Molen et al., 2025)). The trees are now



widely spaced with some open spots. In a radius of 500 m around the flux tower 89% of the area is covered with Scots pine (*Pinus sylvestris*), 3.3% with Corsican or black pine, 2.3% with birch, 1.3% with

Douglas fir, 0.6% with oak (*Quercus Robur*) and 3.5% of the area is open and mostly covered with heather and grass (Moors, 2012). The average tree height increased from 15.3 m in 1996 to 20.6 m in 2020, with a mean annual growth rate of 0.22 m. Trees on top of dunes tend to be shorter than trees growing in the valleys, hence the local topography is not visible from above the canopy. The undergrowth of the forest has exhibited a notable increase in coverage over time, particularly since 1976. It consists of

mosses (*Polytrichum* spp.), grasses (*Deschampsia flexuosa*), blueberries (*Vaccinium myrtilus*), and shrubs (dominated by American cherry of *Prunus serotina* and *Amelanchier lamarckii*) (Moors, 2012) (please refer to photos in Appendix 0). It is noteworthy that at the outset of the observation period, *Vaccinium* was absent, yet it now constitutes the majority of shrubs, indicating a notable increase in its spread over time (please refer to photos in Appendix 0). Because of the local topography caused by the sand dunes, the

distance to the ground water table depends on the location. In the valleys, the ground water table is at a depth ranging from 2.5 to 4.3 m below the surface. More details on the soil and vegetation composition in the first period of the site can be found in the report by Moors (2012), while Van Der Molen et al. (2025) offer insights into Loobos's geological history and ecosystem composition.

The collected dataset contains measurements of meteorological (e.g., radiation components, precipitation,

vertical profiles of wind, air temperature, humidity and $CO_2$, soil temperature and moisture content), turbulent fluxes (i.e., latent heat and sensible heat and $CO_2$) at half-hourly intervals. The dataset from the first tower has been used in many national and international studies and has been cited more than 150 times in peer-reviewed articles, including papers in high impact journals like Nature (Keenan et al., 2013; Valentini, 2000; Enquist et al., 2003). The conducted studies range from: (1) the development of data

quality control and gap-filling strategies for long term energy flux datasets (Falge, 2001; Falge et al., 2001; Reichstein et al., 2005; Pastorello et al., 2020; Meesters et al., 2012; Göckede et al., 2008), (2) data analysis including trend analysis (Dolman et al., 2002; Falge et al., 2002; Tong et al., 2023; Elbers et al., 2011) and to study the response of the ecosystem to droughts, heat waves and warm winters (Lansu et al., 2020; Van Der Horst et al., 2019; Granier et al., 2007; Zhou et al., 2024; Mallick et al., 2024; García-

García et al., 2023; Vermeulen et al., 2015), (3) analyses of carbon and water fluxes exchange dynamics via model development and validation studies (Kramer et al., 2002; Veroustraete et al., 2002; Falge et al., 2003; Papale and Valentini, 2003; Hari et al., 2018; Aubinet et al., 1999; Wu et al., 2020; Strebel et al., 2023; Vermeulen et al., 2015), (4) the development of land surface models and parameter optimizations (Chen et al., 2016; Raoult et al., 2016; Harper et al., 2016; Largeron et al., 2018),  (5) the development of

the CarboEurope regional experiment strategy (Dolman et al., 2006), (6) ecological and land management studies (Van Wijk and Bouten, 1999; Dolman et al., 2003; Ceulemans et al., 2003; Balzarolo et al., 2016;



Churkina et al., 2003; Jansen et al., 2023; George et al., 2021), (7) serving as ground truth for satellite data retrievals such as evapotranspiration (Verstraeten et al., 2005; Hu et al., 2017; Petropoulos, 2024) and gross primary productivity (Verma et al., 2014; Joiner et al., 2014), and (8) groundwater management (Moors, 2012).


While the tower and its associated dataset were described in a limited manner (Dolman et al., 2002; Elbers et al., 2011), the purpose of this paper is to provide a comprehensive overview of the instrumentation, data processing and the resulting data archive, enabling its use in data analysis studies, model development and validation of satellite data retrievals. Sect. 2 describes the instrumentation, basic

data processing, data quality control and obtained data records. Sect. 3 shows data evaluations by presenting data cross-check results, the mean diurnal storage flux in comparison to EC measured fluxes and total fluxes, seasonal and interannual variations in net ecosystem exchange of $CO_2$ flux and energy balance residual. Sect. 4 provides conclusions and information on data and code accessibilities.

## 2 Instrumentation and data processing

### 2.1 In situ measurements

#### 2.1.1 Meteorological variables

At the top of the scaffolding tower (highest platform at 22 m), standard meteorological measurements were conducted, including air temperature, relative humidity, horizontal wind speed and wind direction. Air pressure was measured at the site at 15 m height starting from February 1999. The four radiation

components were measured individually: incoming and reflected shortwave (solar) radiation using two pyranometers, and incoming and emitted longwave (thermal infrared) radiation using two pyrgeometer equipped with a ventilated sensor to ensure reliable readings, particularly during dew and frost events. Additionally, quantum sensors were installed on 9 August 2001 to measure direct photosynthetically active radiation (PAR), diffuse PAR and reflected PAR. These data were sampled every 20 seconds and stored as

half hourly means and standard deviations.

Precipitation was measured using tipping bucket rain gauges with a resolution of approximately 0.2 mm per tip. One rain gauge was located on top of the tower. Another rain gauge was installed in an open space nearby to minimize the error due to high wind. However, this open space measurement was discontinued after 9 January 2007 due to the regrowth of pine trees, which caused the area to no longer be open

enough. Throughfall was measured from 2 June 1995 until 23 July 2014 using 36 manual gauges as well as a custom made tipping bucket rain gauge at the end of an approximately 10 meters long gutter through with a width of 10 cm. The manual gauges were set up at a 4 m distance from each other in a fixed square of 400 $m^2$. The area around these manual gauges was kept free of grass and shrubs. The resolution of the





tipping bucket gauge used for the throughfall trough is approximately 0.07 mm per tip depending on the
exact surface of the gutter and the precipitation density. This tipping bucket, along with the one on top of
the tower, was initially logged at a 5 minute intervals, with the tips accumulated over each interval. Since
16 June 2004, a Campbell logger has been recording data at 30 minutes intervals. Detailed information
about the instruments, their manufacturers and their specific locations on the tower is provided in Table 1.

*Table 1 List of instruments, installation height and measurement period.*

| Variable | Instrument | Manufacturer | Type | Height  above ground (m) | Measurement period |
|---|---|---|---|---|---|
| Incoming/reflected short wave radiation | Pyranometer | Kipp&Zonen | CM21 | 21.9 | 1995-Jan-05 to 2023-Mar-30 |
| Incoming/reflected long wave radiation | Pyranometer | Kipp&Zonen | CG1 | 21.9 | 1995-Jan-05 to 2023-Mar-30 |
| Temperature longwave radiation sensors | Platinum resistance | Kipp&Zonen | PT100 | 21.9 | 1995-Jan-05 to 2023-Mar-30 |
| Air temperature | Platinum resistance | Vaisala | HMP35A | 23.5, 7.5, 5.0 | 1995-Jan-05 to 2023-Mar-30 |
| Air pressure | Analog barometer | Vaisala | PTB101C | 15 | 1996-July-22 to 2023-Mar-30 |
| Relative humidity | Capacitive sensor | Vaisala | HMP35A | 23.5, 7.5, 5.0 | 1995-Jan-05 to 2022-Dec-31 |
| Wind speed | Cup anemometer | Vector Instruments | A101ML | 24.4, 7.5, 5.0 | 1995-Jan-05 to 2022-Dec-31 |
| Wind direction | Wind vane | Vector Instruments | W200P | 24 | 1995-Jan-05 to 2022-Dec-31 |
| Precipitation | Tipping bucket | EML | ARG100 | 23.9 (tower), 0.4 (open field) | 1995-Jan-05 to 2023-Mar-30 |
| Throughfall | Tipping bucket | IMAG-DLO | - | 1 | 1995-June-02 to 2014-July-23 |
| Direct photosynthetically active radiation | Quantum sensor | Delta-T Devices | BF-3 | 24.5 | 2001-Aug-09 to 2023-Mar-30 |
| Diffuse photosynthetically active radiation | Quantum sensor | Delta-T Devices | BF-3 | 24.5 | 2001-Aug-09 to 2023-Mar-30 |
| Reflected photosynthetically active radiation | Quantum sensor | LI-COR | LI-190SZ | 21.9 | 2001-Aug-09 to 2023-Mar-30 |
| | A single channel infrared | PP Systems, | CIRAS-SC | 25.97, 23.22, 7.5, 5.0 and 2.5 m before October 1999, and | 1996-July-23 to 2007-Oct |



| Measurement | Instrument type | Manufacturer | Model | Height/Depth | Period |
|---|---|---|---|---|---|
| CO$_2$ mole fraction and H$_2$O pressure | gas analyser (IRGA) | | modified NOAA system | 25.97, 7.5, 5.0, 2.5 and 0.4m afterwards. | |
| | AIRCOA: Autonomous Inexpensive Robust CO2 Analyzer | LI-COR | Li-Cor LI-820 | 25.97, 7.5, 5.0, 2.5 and 0.4m | 2007-Oct to 2021-Nov-12 |
| Turbulence components | Sonic anemometer | Gill instruments | A Gill Solent 1012R2 sonic anemometer | 27 | 1996-Aug-21 to 2001-June-07 |
| | Sonic anemometer | Gill instruments | Windmaster Pro | 27 | 2001-June-07 to 2016-June-01 |
| | Sonic anemometer | Gill instruments | Gill R3-50 ultrasonic anemometer | 27 | 2016-June-01 to 2022-Oct-11 |
| CO$_2$/H$_2$O fluctuations | KH2O | | Krypton hygrometer | 27 | 1997-Jan-08 to 2001-June |
| | Infrared gas analyser | LI-COR | Li-Cor LI-6262 | 27 | 1997-Jan-08 to 2001-June |
| | | | LI-7500 | 27 | 2001-June to 2019-May |
| | | | Li-COR LI-7500A | 27 | 2019-May to 2019-Aug-08 |
| | | | LI-7500RS | 27 | 2019-Aug-08 to 2022-Oct-11 |
| Soil moisture and temprature | FD sensor and thermistor | MUXCOM | Frequency domain (FD) | litter, -0.03, -0.20, -0.50, -1.0 | 1995-Mar-01 to 2000-Sep-14 |
| Soil moisture | FD sensor | Campbell Scientific | CS616 | litter, -0.03, -0.20, -0.50, -1.0 | 2005-Apr-11 to 2023-May-31 |
| Soil temperature | Thermistor | Campbell Scientific | 107 | litter, -0.03, -0.20, -0.50, -1.0 | 2005-Apr-11 to 2023-May-31 |
| Soil heat flux | Thermopiles | Hukseflux | Similar to FHF05 series heat flux sensor. | -0.1 | 1995-Jan-05 to 1998-Jan |
| | Thermopiles | TNO-TPD, SH1 | PU43T, Hukseflux | -0.1 | 1998-Nov to 2017-Sep |
| | Thermopiles | Campbell instruments | HFP01SC | -0.1 | 2017-Sep to 2023-Mar-18 |
| Soil respiration | Infrared gas analyser | PP Systems | EGM-4 | -0.15 | 2001-Jun-28 to 2010-July-22 |
| Leaf area index | Plant canopy analyzer | LI-COR | LAI-2000 | | 1996-May-22 to 2014-07-23 |
| Bole temperature | Thermistor | Campbell Scientific | 107 | 4 | 2005-Apr-11 to 2023-May-31 |





| Sap flow | | | | | |
|---|---|---|---|---|---|
| | Čermák system | Ecological Measuring Systems (Brno, Czech Republic) | Model P4.1 | 15 | 1996-July-05 to 1998-Aug-16 |
| | Thermal dissipation probes | Dynamax | Model P4.1 | 17 | 2011-Jan-01 to 2015-Nov-11 |
| Photosynthesis measurements (e.g., light response curve) | Intelligent Photosynthesis System | ADC Bioscientific Ltd. | LCpro-SD | 15 | 1997-July/Aug/Nov, 1998-July/Aug, 2000-July |
| Groundwater level (filter depth) | Tube | Manual | | -6.5, -4.8 | 1995-Jan-01 to 2018-Dec-19 |


### 2.1.2 Eddy covariance

An EC-system placed on the top of the tower at 27 m was used to measure turbulent fluxes (i.e., sensible and latent heat fluxes and $CO_2$ flux). The measuring system involved a 3D ultrasonic anemometer and a fast infrared gas analyser. In the first setup (from fall 1996 till June 2001) a Gill R2 was used in

combination with a Li-COR LI-6262 and a Campbell Krypton hygrometer $KH_2O$ (Table 1). Raw data were stored at 10 Hz using a HP Palmtop PC and PCMCIA cards. Since June 2001 a Windmaster Pro anemometer was installed in combination with an open path Li-COR LI-7500 (Table 1). The Li-COR LI-7500 was replaced by Li-COR LI-7500A in May 2019, and subsequently, on 8 August 2019 by LI-7500RS (Table 1). The Windmaster Pro anemometer was replaced by a Gill R3-50 ultrasonic anemometer

on 1 June 2016 (Table 1).

### 2.1.3 Below canopy profile of $CO_2$ mole fraction, $H_2O$ pressure and temperature

Together with turbulent flux measurements, a single channel infrared gas analyser (CIRAS-SC, PP Systems) and a solenoid switching system were deployed to measure $CO_2$ mole fraction at five levels above ground (25.97, 23.22, 7.5, 5.0, 2.5 m before 1[st] October 1999, and 25.97, 7.5, 5.0, 2.5, 0.4 m

afterward, Table 1) in and above the canopy (Dolman et al., 2002). After October 2007 an Autonomous Inexpensive Robust $CO_2$ Analyzer (AIRCOA, NOAA) system (Stephens et al., 2006; Stephens et al., 2011) was deployed to measure profile $CO_2$ and $H_2O$ model fraction (Elbers et al., 2011). Compared to the original system built for three levels, the system at the site was adjusted to sample gas model fraction at five levels (25.97, 7.5, 5.0, 2.5, 0.4 m, Table 1). The AIRCOA system was composed of: (1)  a gas

sampling system and a gas flow control system that regulates the alternating of calibration gas (H2, H1, L1, L2 and LT in Fig. 1) and ambient air (the black rectangle labelled 3 in Fig. 1) and the periodic calibration of the system; (2) An Infrared Gas Analyzer (IRGA, Li-COR LI-820) that measures the model



fraction of $CO_2$ using infrared absorption techniques; (3) A filtering (the black rectangle labelled 5 and 40 in Fig. 1) and drying system with nafion tubes and molecular and moisture sieves for obtaining clean and dry samples for IRGA analysis; (4) a section of CPU, DAQ (data acquisition), IO (input/output) power in a PC-based computer for performing automated data acquisition and valve control (Fig. 1). By alternating between ambient air and gas from cylinders containing calibration gases that were free from particulates and water vapor, $CO_2$ model fraction were measured by the IRGA and the IRGA was automatically calibrated on a daily basis. To obtain profiles of water vapor pressure, the relative humidity and temperature for sampled moist air were measured before entering the drying system (RH/T measured before 7th).

The IRGA datalogger was configured to record raw data in two seconds interval. Regarding the measurement accuracy, the two second filtered values exhibited one standard deviation root mean square error of 0.6 ppm, which averaged to 0.1 ppm over 100 seconds. The absolute accuracy of the $CO_2$ mole fraction measurements with the AIRCOA system was 0.2 ppm higher than 2 ppm with the CIRAS system (Elbers et al., 2011). The instrument switched the gas being analysed every 160 seconds in this case. Every 4 hours the instrument measured all four calibration gases to obtain an estimate of the calibration coefficients for the IRGA, and every 8 hours the instrument analysed the long-term surveillance gas (LT in Fig. 1). The $CO_2$ model fractions in five calibration cylinders provided by University of Groningen were steadily maintained throughout the whole period and the corresponding collected calibration data are listed in Table A1 in Appendix A for reference. Additionally, air temperature, wind speed and relative humidity measurements were collected at two more levels below the canopy (7.5 and 5.0 m, Table 1).

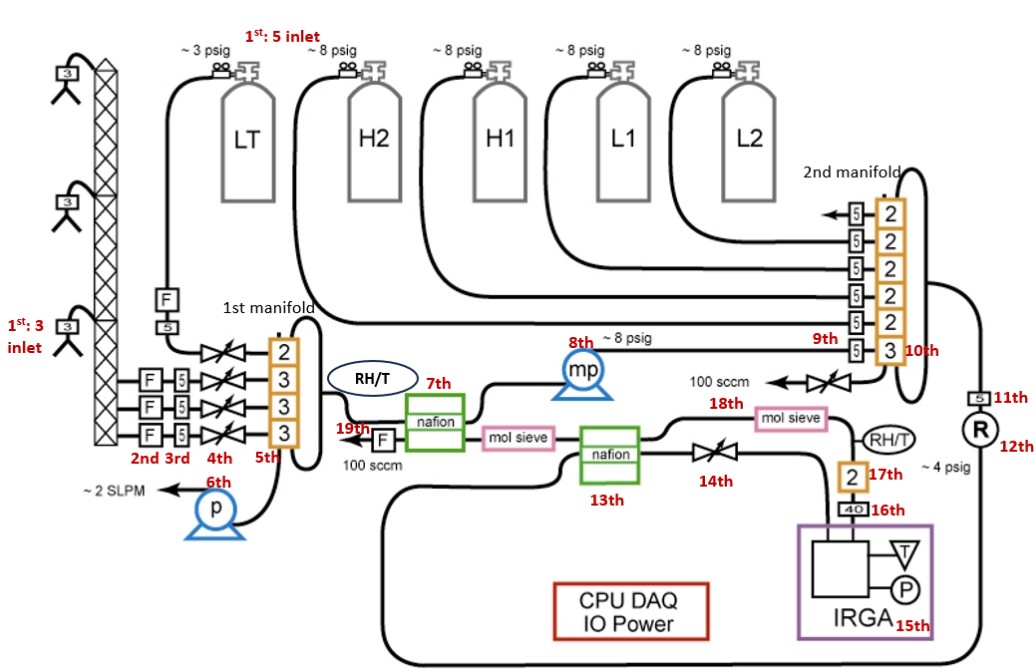

Figure 1 Schematic diagram of the 5-calibration inlet and 5-sample inlet AIRCOA system that was operational in the Loobos first tower. This figure is adapted from Stephens et al. (2006) and only 3 inlets are shown for simplicity. The 5 calibration inlets are taken from the H2, H1, L1, L2 and LT cylinders, where LT stores a long-term surveillance gas for verifying the other four calibration gases. The 5 sample inlets are deployed at different levels in the tower. Each inlet stream ($1^{st}$) passes through a mass flow meter F ($2^{nd}$), and a 5 μm metal filter labelled 5 in the following ($3^{rd}$) and a needle valve ($4^{th}$) before reaching a manifold of three-way (3) and two-way (2) solenoid valves ($5^{th}$). A diaphragm pump labelled p ($6^{th}$) in the blue circle flushes the sample lines to modulate the flow rate (e.g., 2 SLPM (standard liters per minute)) and system pressure. The gas selected by these valves passes through the Nafion driers ($7^{th}$), and a smaller diaphragm pump in the blue circle ($8^{th}$) is used to compress the dry gas to increase pressure (e.g., into 8 psig). Then the gas passes through a second 5 μm metal filter ($9^{th}$) and goes into a second manifold of three-way (3) and two-way (2) solenoid valves ($10^{th}$). The second manifold selects either a sample gas or a calibration gas for analysis. The select gas then passes through another 5 μm metal filter ($11^{th}$) and a miniature pressure regulator R in bold ($12^{th}$). The gas next is dried by a Nafion drier ($13^{th}$) and reduced in pressure by a needle valve ($14^{th}$), which is normally used to adjust the sample flow to 100 sccm (standard cubic centimeters per minute). The gas is then analyzed by a LI-820 Infrared Gas Analyzer for measuring $CO_2$ model fraction and pressure and temperature (T and P) as well ($15^{th}$). After leaving the IRGA, the gas goes through a metal filter of 40 μm ($16^{th}$) and a valve used for leak check purposes, and a humidity and temperature sensor (RH/T) to verify drier performance ($17^{th}$). The gas is further completely dried by molecular sieve ($18^{th}$), and then goes through a final mass-flow meter ($19^{th}$) followed by exhausting to the atmosphere at the end.

### 2.1.4    Soil properties, soil heat flux, profile soil moisture and soil temperature and soil respiration

Soil heat fluxes were measured by four thermopiles under the litter layer at a depth of 3 cm in the mineral soil at a total depth of 10 cm. Thermopile sensors arranged in a thin ring (similar in design to FHF05 series heat flux sensors, https://www.hukseflux.com/products/heat-flux-sensors/heat-flux-sensors/fhf05-



series-heat-flux-sensorss, last access: 15 December 2024) built by Hukseflux were used. After serious lightning damage to the sensors in January 1998, two new sensors (TNO, PU43T and SH1-Hukseflux thermal sensors, Table 1) were placed in November 1998 and remained operational until September 2017.

This sensor was located in between the two soil moisture profiles (described below) directly on the mineral soil under the litter layer. Since September 2017 the HFP01SC soil heat flux plate (https://www.campbellsci.com/hfp01sc-l, last access: 15 December 2024) was deployed near the location for measurements until 18 March 2023.

A change of systems throughout the long measuring period occurred as well for soil temperature,
electrical conductivity and soil moisture measurements, which were initially measured at five different depths in two profiles 1.5 to 2.0 m apart. The MUXCOM (IMAG-DLO, Multiplexed Control and Monitoring) system containing frequency domain sensors at the 20 MHz frequency range was deployed in two profiles for measurements until 14 September 2000. Every 30 minutes a measurement was made at all sensors and stored on a palmtop PC. On 11 April 2005 Campbell water content reflectometer sensors
CS616 at the 70 MHz frequency range were deployed in one profile and remained operational until 31 May 2023 (Table 1). The Campbell sensors were logged by a Campbell logger that recorded all soil measurements at 30 minutes intervals. To obtain an accurate estimation of the soil moisture content, calibration curves were made using undisturbed soil samples with a diameter of 20 cm and a height of 20 cm taken at different depths.

Soil respiration measurements were conducted from 2001 to 2010 along one transect with 22 sampling points, extending from the tower to an open area (at the time of measurement). The transect included both updune and lowdune locations. At each point, the soil respiration chamber was inserted into the soil with a depth of 15 cm and the grass within the chamber (i.e., SRC-1 soil respiration chamber in this case) was cut if necessary, in order to exclude photosynthesis and plant respiration measurements. Soil $CO_2$ fluxes
were measured using an EGM-4 infrared gas analyser (PP Systems, Table 1) with built-in soil temperature sensors. Soil moisture was simultaneously measured with Theta Probes (Delta-T Devices).

### 2.1.5    Vegetation properties

#### Tree inventory

Vegetation properties such as tree diameter at breast height and tree height were measured from 1996 to
2012. The main tree species is Pinus sylvestris (Moors, 2012), given the measured tree diameter and height, the above ground biomass (Table ) was estimated by using the allometric relations (please refer to Schelhaas et al. (2022) and section 1.2.4 in Van Der Molen et al. (2025)).

#### Leaf Area Index (LAI)



The Leaf Area Index (LAI) was measured regularly between 1996 and 2014 with two LAI-2000 (Li-
COR) simultaneously. Measurements were typically conducted biweekly during the growing season and
monthly otherwise. One instrument was mounted atop the scaffolding tower providing a reference
measurement of incoming light. Meanwhile, the other was used along the transect below the canopy
measuring light attenuation, as such, LAI estimation was made. The setup consisted of 70-100
measurement points with a 3 m spacing below the canopy to provide better spatial representation. The
LAI-2000 sensor measurements were calibrated by comparing them with results from destructive
sampling (Moors, 2012). The LAI data are listed as ancillary data. Additionally, the Campbell thermistor
was deployed since 11 April 2005 to measure bole temperature.

### Sap flow

At the Loobos site sap flow was measured with a Tissue Heat Balance-system of Čermák (Ecological
Measuring System, model P4.1, Brno, Czech Republic) from 1996 to 1998. By measuring temperature
changes of the phloem, the amount of energy needed for heating and the specific heat of water, the sap flow
was calculated without the necessity of calibrations (Lundblad et al., 2001). Detailed information can be
found in Moors (2012).

Between 2012 and 2015 the sap flow was measured with thermal dissipation probes (Dynamax, Table 1)
based on the temperature difference between the heated needle and the sapwood ambient temperature.
Sapflux was calculated following Granier (1987). The supplied data were averages of two sensors deployed
at six trees. The data gaps were mainly due to power shortages and mainly during nights and winter.

### Needle foliage properties

A number of needle leaves from trees around the tower were collected to measure needle foliage area, dry
weight and leaf mass per area. The foliage area was determined using image analysis software
(https://imagej.net/ij/), the dry weight was obtained after oven-drying at 60°C, and the leaf mass per area
was calculated as the ratio of dry weight to foliage area. Total carbon (C) and nitrogen (N) concentrations
were determined using dry combustion of ground plant material with a CHNS/O elemental analyzer
(PerkinElmer 2400 Series II). Total phosphorus (P) concentration was measured by digesting ground leaf
material in 37% hydrochloric acid (HCL) followed by colorimetric measurement at 880 nm after reacting
with molybdenum blue.

Photosynthesis measurements involving the light response curve, $CO_2$ response curve and the daily and
seasonal responses of photosynthesis were conducted with an intelligent portable photosynthesis system
(ADC Bioscientific Ltd., Table 1). The measurements between 1997 and 1998 were performed on the top
of the tower for sun-exposed leaves, and the measurements in 2000 were performed on the top of a tree
randomly selected in the north of the tower. The experiments of obtaining light and $CO_2$ response curves



were conducted for two summer days in 1997 (Aug-07/18) and 2000 (July-19/20), and daily response
measurements were on 1998-Aug-11. The seasonal response of photosynthesis was measured on 1997-
July-29, 1997-Aug-04/11/21, 1997-Nov-22, 1998-July-22. Data of $CO_2$ assimilation rate, transpiration
rate and stomatal conductance at the leaf level were obtained.

### 2.1.6 Ground water level

The data of the ground water level (GWL) were measured manually in two observing tubes (2.5 cm
diameter). The error in the measurements is less than 1 cm. Due to the dunes there is a distinct local
topography with a variation in height of about 2 m in the immediate surroundings of the tower, but at
distances more than 100 m away, the dunes may reach 10 m above the valleys. It is unknown how this
orography influences the GWL. Data presented as ancillary are from a tube (B15) +/- 30 m northeast of the
flux tower in a local valley +/-2.7 m below the base of the flux tower.

## 2.2    Data processing

The general processing pipeline for continuous recorded datasets is schematically shown in Fig. 3. The
recorded meteorological raw data was imported into a Paradox relational database management system
and categorized into two streams: NL-Loo_BM and NL-Loo_Profile, where NL refers to the Netherlands
and Loo denotes the site name.

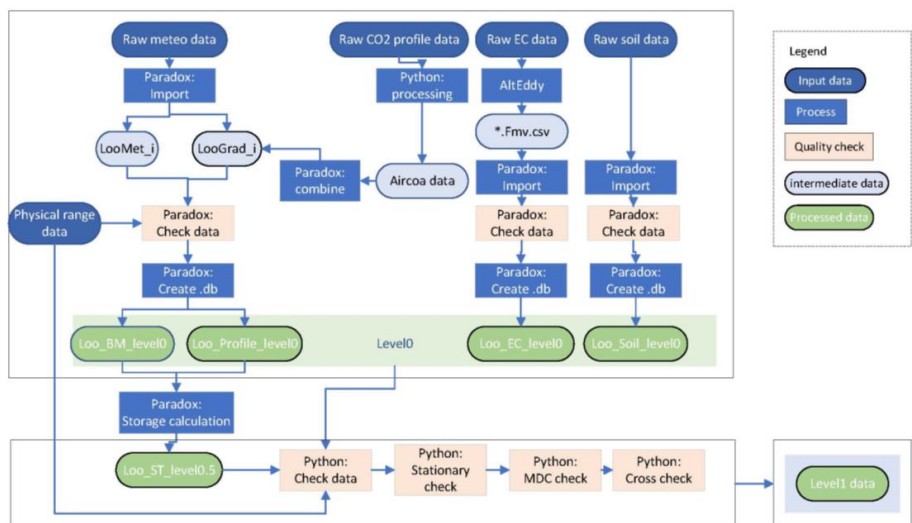

*Figure 2 The workflow for processing data from the first tower at Loobos.*





The NL-Loo_BM stream contains fundamental meteorological variables, while the NL-Loo_Profile stream primarily includes profile $CO_2$ and $H_2O$ pressure data, which were derived from processing PP system and AIRCOA system measurements as described in Sect. 2.2.1. Following this, heat storage and fluxes beneath the canopy were calculated as described in Sect. 2.2.2. The recorded raw EC data were processed using AltEddy software to estimate fluxes (Elbers et al., 2011), as described in Sect. 2.2.3. The

resulting flux data, along with the recorded raw soil moisture and temperature data were also imported into the Paradox database. Data quality was verified using predefined physical ranges (Table A2 in Appendix E), and the data was further completed on a yearly scale by replacing missing data with NA values to ensure consistency across all datasets (Fig. 2). By combining meteorology, storage, EC and soil data, the net ecosystem exchange (NEE) rate of $CO_2$, latent heat flux (LE) and sensible heat flux (H) were

computed with NEE being gap-filled, as described in Sect. 2.2.4. In total, four level 0 data streams—NL-Loo_BM, NL-Loo_Profile , NL-Loo_EC and NL-Loo_Soil–were obtained (Fig. 2), together with a level 0.5 data stream of NL-Loo_ST that were derived from the Level 0 dataset. Level 0 implies raw data, as observed and/or based on basic computations. Level 0 and Level 0.5 data which undergo extensive quality control are lifted to Level 1 data.

Additionally, the datasets of soil respiration, vegetation properties (i.e., tree height, stem width and dry aboveground biomass, Leaf Area Index, sap flow, needle foliage properties and the associated nutrient analysis, and photosynthesis response curves) and ground water level are stored as ancillary data.

### 2.2.1     *Calculation of profile $CO_2$ mole fraction and $H_2O$ pressure*

Using the recorded raw data at two-second intervals and the collected calibration data, $CO_2$ mole fractions

at the five altitude levels were calculated, as illustrated in Fig. 3. The main procedures included (1) Screening the line data to match the appropriate level, (2) Calculating median values for each data level over a time block of 160 seconds, corresponding to the gas switch frequency in this case, (3) incorporating calibration data into each block, (4) Computing calibration curves for each eight hour cycle based on LT operation cycle, (5) Applying calibration coefficients to the level data at 15-minute intervals.

The calibration effects can be viewed in Figs. A2 and A3 in Appendix C.



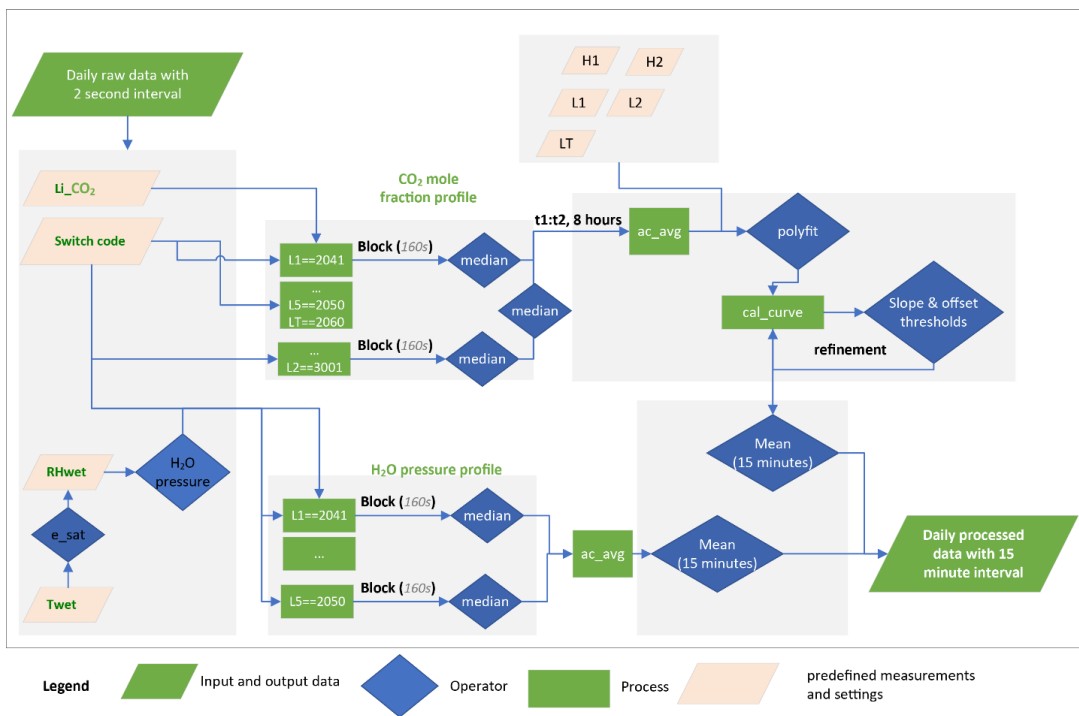

*Figure 3 Flowchart for processing profile $CO_2$ mole fraction (ppm) and $H_2O$ pressure (mbar) measured by AIRCOA.*

The water vapor pressure at the five levels was derived from the measured relative humidity and

temperature of the sampled gas before it entered the drying system (see Fig. 3), using standard equations (Eqs. (A1-A2) in Appendix B). Since $H_2O$ sensors are stable and less affected by interferences from other gases and environmental factors, no additional calibration was required, unlike the $CO_2$ mole fraction measurements. The same aggregation method was applied to obtain the profile of the $H_2O$ model fraction at 15-minute intervals. The flowchart illustrating the calculation process for the profile $CO_2$ and $H_2O$

model fraction is presented in Fig. 3.

### 2.2.2   Calculation of heat storage and fluxes beneath the canopy

Sensible fluxes of sensible heat beneath the canopy were derived from the measured air temperature at the three levels, following the equations outlined in Appendix D (Eqs. (A3-A4)). In a similar manner, $CO_2$ and $H_2O$ storage fluxes were derived using the measured $CO_2$ mole fraction and water vapor pressure at

five levels. The storage data was saved for use in the NL-Loo_ST _level0.5 stream.



### 2.2.3 Estimate of turbulent fluxes

The turbulent fluxes are estimated as covariances between vertical wind speed and the scalar quantities of interest (heat, water vapor, $CO_2$). To derive these flux estimates from the raw EC measurements, the AltEddy software (Version 3.90, from Wageningen Environmental Research (WEnR), The Netherlands) was used. This software executes a series of essential processing steps, including detrending, time-lag correction, double coordinate rotation correction to align wind velocity components with the mean wind direction, angle of attack correction, density fluctuation compensation (Webb et al., 1980), normalized spectra and cospectra calculations. Detailed information about AltEddy software can be found at https://www.climatexchange.nl/projects/alteddy/, see also Mauder (2008), which demonstrated the capability of the AltEddy software in calculating $CO_2$ fluxes compared to those calculated by other software packages. The output data saved in the NL-Loo_EC_level0 stream contains the turbulent and $CO_2$ fluxes and the corresponding quality flag from Foken et al. (2004), as well as the means of wind and scalars and the turbulent and flux parameters including friction velocity ($u^*$), stability parameter $z/L$, wind direction and the 80% distance integration of the flux derived from Schuepp et al. (1990).

## 2.3 Data quality control

Physical data ranges, including maximum and minimum values, maximum and minimum values of differences between consecutive time steps, and the maximum standard deviation of the interest of field, were applied to filter out abnormal values and missing data from level 0 NL-Loo_BM, NL-Loo_Profile and NL-Loo_Soil streams, as illustrated in Fig. 2 (light orange box). Table A2 shows the maximum and minimum physical ranges, maximum differences, and maximum standard deviations applied for variables.

Regarding the turbulent fluxes and $CO_2$ flux from the NL-Loo_EC stream, an initial quality flag was produced by running the AltEddy software. The data quality was further refined (Fig. 2). Specifically, flux subset data were created and used for the determination of the response of daytime flux to incoming solar radiation and of nighttime flux to air temperature. Subsequently, flux data were discarded when they fell outside tolerable ranges (a bin-average ± 2 standard deviation), and the corresponding quality flag data were reassigned accordingly. Detailed descriptions can be found in Elbers et al. (2011).

The level 0 and level 0.5 data was further reviewed and refined into consistent level 1 data through four procedures (1) reapplying physical range criteria (Table A2), (2) filtering out stationary data within a three-hour window (with the exception of the NL-Loo_Soil stream and shortwave radiation in the NL-Loo_BM stream), (3) calculating the long-term mean diurnal cycle (MDC) and its standard deviations for $CO_2$ flux and accordingly filtering out daytime data not in the tolerable range, and (4) comparing similar fields from different streams for cross-checks.

We thus provide a level 1 data product, which consists of quality controlled measured variables and derived variables (e.g. eddy fluxes and storage fluxes). Here we do not provide gap filled data, since the gap-filling will be done centrally by ICOS/FLUXNET in a homogenised way. We anticipate that the gap-filled data will become available via ICOS and FLUXNET as part of the net FLUXNET Data System by December 2025.

## 2.4 Period of record

Please refer to the description Excel sheet to review the variables included in each data stream. The Level 1 data were stored in one CSV file per stream, where the variable name is consistent with ICOS standards. The ancillary data were stored in Excel per stream. Level 1 data availability is presented in both Fig. 4 and Table 1.

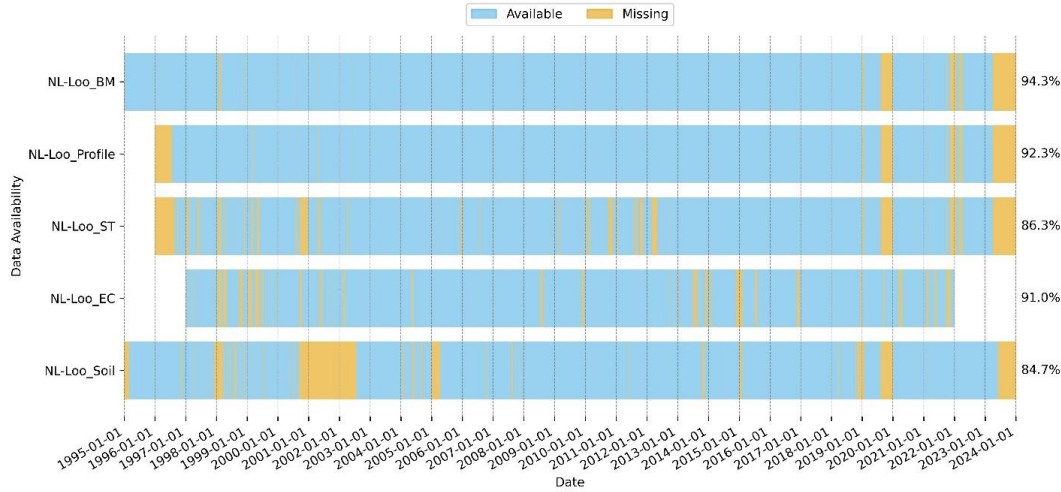

*Figure 4 The availability of the five continuous data streams. NL-Loo denotes the origin of the site. The NL-Loo_BM stream includes biometeorological data, the NL-Loo_Profile stream contains vertical profile data, the NL-Loo_ST stream includes storage data, the NL-Loo_EC stream includes EC measurement data, and the NL-Loo_Soil stream includes soil moisture and temperature data.*

## 3 Data evaluation

### 3.1 Data cross-checks

To validate the meteorological data measured by various sensors, statistical metrics including Pearson's correlation coefficient (R), mean bias (Bias) and root mean square error (RMSE) were calculated for air





temperature (TA) measured by Vaisala against sonic temperature, profile wind speed (WS) measured by
cup anemometer against WS from sonic, and wind direction (WD) measured by the wind vane and sonic

against WD from KNMI meteorological station at Deelen (https://www.knmi.nl/nederland-
nu/klimatologie/daggegevens), the closest KNMI station to the Loobos first tower. Table 2 shows high R
values for temperature and WS measured by different sensors. Compared to WD measured by sonic, the
WD measured by the wind vane exhibits higher R and lower RMSE when compared with the KNMI daily
dataset. The use of WS and WD from the wind vane is thus recommended for further analysis. Table 3

shows decent correlation coefficients (> 0.5) between $CO_2$ mole fraction and $H_2O$ pressure measurements
from the AIRCOA system and those from the EC system. These cross-check results demonstrate the
consistency of datasets measured by different sensors.

*Table 2 Statistics comparing meteorological data measured by different sensors from 1997 to 2022. TA is expressed in °C, WS is in unit of m/s and WD in degrees.*

| Name | R | Bias | RMSE |
|---|---|---|---|
| TA (Vaisala vs. sonic) | 0.88 | -3.4 | 4.7 |
| WS_24.4 m (cup anemometer vs. sonic) | 0.91 | -0.1 | 0.5 |
| WS_7.5m (cup anemometer vs. sonic) | 0.80 | -2.2 | 2.3 |
| WS_5.0m (cup anemometer vs. sonic) | 0.79 | -2.2 | 2.3 |
| WD (vane vs. KNMI) | 0.78 | -11.5 | 56.5 |
| WD (sonic vs. KNMI) | 0.21 | -2.9 | 102.1 |
| WD (sonic vs. vane) | 0.26 | 6.3 | 87.1 |


*Table 3 Statistics comparing profile $CO_2$ mole fraction and $H_2O$ pressure data measured by different sensors at different altitude levels (from 1 to 5 representing 24.4, 7.5, 5.0, 2.5 and 0.4 m) in the canopy. $CO_2$ is expressed in unit of ppm and $H_2O$ in mbar.*

| Name | R | Bias | RMSE |
|---|---|---|---|
| $CO_2$_1_1_1 & $CO_2$_2_1_1 | 0.68 | -0.9 | 23.4 |
| $CO_2$_1_1_1 & $CO_2$_2_2_1 | 0.63 | -3.7 | 25.7 |
| $CO_2$_1_1_1 & $CO_2$_2_3_1 | 0.62 | -5.0 | 26.8 |
| $CO_2$_1_1_1 & $CO_2$_2_4_1 | 0.59 | -7.8 | 29.3 |
| $CO_2$_1_1_1 & $CO_2$_2_5_1 | 0.56 | -10.5 | 32.0 |
| $H_2O$_1_1_1 & $H_2O$_2_1_1 | 0.58 | -0.8 | 3.7 |
| $H_2O$_1_1_1 & $H_2O$_2_2_1 | 0.57 | -0.8 | 3.7 |
| $H_2O$_1_1_1 & $H_2O$_2_3_1 | 0.59 | -0.8 | 3.7 |
| $H_2O$_1_1_1 & $H_2O$_2_4_1 | 0.59 | -0.9 | 3.7 |
| $H_2O$_1_1_1 & $H_2O$_2_5_1 | 0.58 | -0.9 | 3.7 |

Regarding vegetation data, the tree height (Table 4) matches well with the tree height measured in the
target area in the current years (Table 3 in Van Der Molen et al. (2025)). Two datasets suggest a growth





rate of 16 cm per year. As the Pinus sylvestris species dominate in the study area (Moors, 2012) during the period from 1996 to 2025, by assuming the average tree density of 499.1 trees ha$^{-1}$ (Van Der Molen et al., 2025), the above ground biomass was estimated by using the allometric relations (please refer to

Schelhaas et al. (2022) and section 1.2.4 in Van Der Molen et al. (2025)) and measured tree height (Table 4) and diameter (Table 5). A larger biomass during this period (Table 6) is observed than that in the 2023 inventory based on over 1000 trees (Van Der Molen et al. (2025)). Nevertheless, the estimated biomass is in the same order of magnitude.

Here the tree density in 1996 is based on an inventory of 150 trees, while only for 56 trees the height and

diameter were measured. Information about the plot size and consequent tree density in 2000 has been lost. By plotting the tree coordinates on a map, it has been derived that the plot size must have been between 45 x 45 m and 50 x 50 m. A written report claims a tree density of 499.1 trees ha$^{-1}$ in 2000 which is consistent with 103 trees in 0.21 ha (45.4 x 45.4 m). This is the plot size we assumed. In subsequent years, the tree density was decreased with the number of fallen trees in the inventory. In 2025 68 of the

103 original tree tags have been found. Others have disappeared or grown into the bark. In addition, 31 untagged trees were found in a 50 x 50 m square around those 68 trees. This implies that the tree density decreased from 465 in 2012 to 396 trees ha$^{-1}$ in 2025, a decrease of 17 trees in the quarter hectare plot in 13 years, which seems realistic considering the number of dead stems observed in the field. However, some stems have completely been decomposed in the meantime.

*Table 4 Tree height measurements between 1996 and 2025.*

| Year | 1996 | 2000 | 2005 | 2008 | 2012 | 2025 |
|---|---|---|---|---|---|---|
| **Average (m)** | 15.3 | 15.8 | 16.7 | 17.6 | 18.6 | 20.5 |
| **Standard error (m)** | 0.3 | 0.2 | 0.2 | 0.3 | 0.3 | 0.3 |
| **Count** | 56 | 103 | 100 | 99 | 98 | 102 |
| **Max (m)** | 22.8 | 22.0 | 21.5 | 24.4 | 28.5 | 26.0 |
| **Min (m)** | 9.0 | 8.7 | 9.0 | 8.0 | 12.0 | 11.0 |

*Table 5 Tree diameter measurements between 1996 and 2025.*

| Year | 1996 | 2000 | 2005 | 2008 | 2012 | 2025 |
|---|---|---|---|---|---|---|
| **Average (cm)** | 26.7 | 27.1 | 28.2 | 28.1 | 29.3 | 33.5 |
| **Standard error (cm)** | 0.6 | 0.5 | 0.6 | 0.6 | 0.6 | 0.7 |
| **Count** | 66 | 103 | 100 | 96 | 97 | 103 |
| **Max (cm)** | 41.6 | 43.0 | 43.0 | 41.7 | 42.5 | 49.7 |
| **Min (cm)** | 13.8 | 14.0 | 13.5 | 13.1 | 13.5 | 18.8 |





*Table 6. Above ground biomass (ton dry matter ha⁻¹) estimated between 1996 and 2025. The 1996 tree density is low relative to 2000. In brackets the 2000 tree density and the resulting total above ground biomass estimate.*


| Year | 1996 | 2000 | 2005 | 2008 | 2012 | 2025 |
|---|---|---|---|---|---|---|
| **Tree count** | 56 | 103 | 100 | 96 | 96 | 99 |
| **Plot size (ha)** | 0.15 | 0.21 | 0.21 | 0.21 | 0.21 | 0.25 |
| **Tree density (trees ha⁻¹)** | 362 (499) | 499 | 485 | 465 | 465 | 396 |
| **Average above ground tree mass (kg tree⁻¹)** | 252 | 243 | 244 | 284 | 325 | 437 |
| **Total above ground biomass (ton ha⁻¹)** | 91 (126) | 121 | 118 | 132 | 151 | 173 |

The resulting LAI is 1.65 on average, but variable over the season (Fig. 5), with an increase after budburst in late spring and early summer, a clear maximum in August and a decline in fall, associated with partial leaf shedding. A distinct inter-annual variation was observed (Fig. 6), although it is unknown what the underlying cause is, we note that storms in 2007 caused trees and tree tops to break and 2003, 2013 to be dry years, 2007 and 2008 to be wet years. Additionally, Table 7 presents comparable needle foliage attributes measured between 1998 and 2012 with those from 2024 (Van Der Molen et al., 2025).


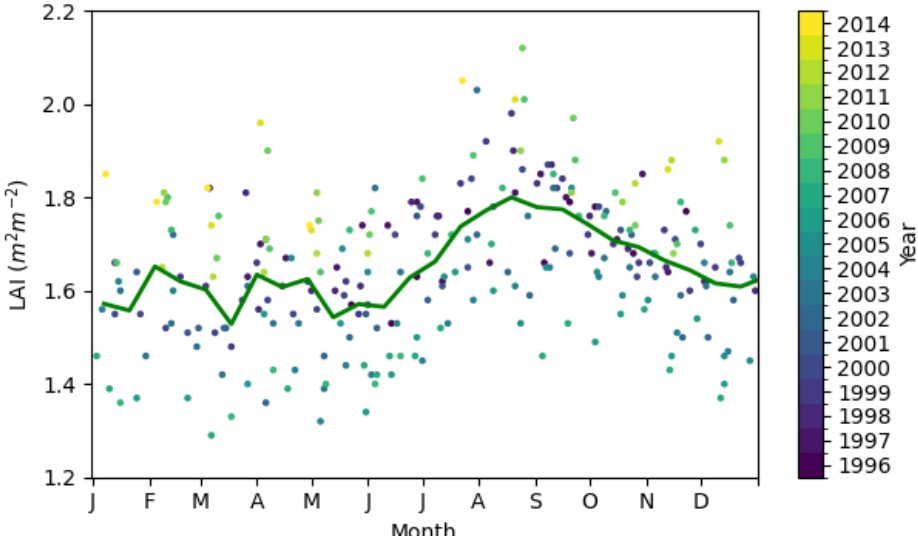

*Figure 5 Seasonal cycle of Leaf Area Index. The colored points indicate the individual measurements and their year of measurements. The solid line shows the 14 day mean. Each data point is the average of 60 samples collected 10 m apart in two 300 m transects crossing at the first tower. The tick marks on the x-axis indicate the first day of the month.*




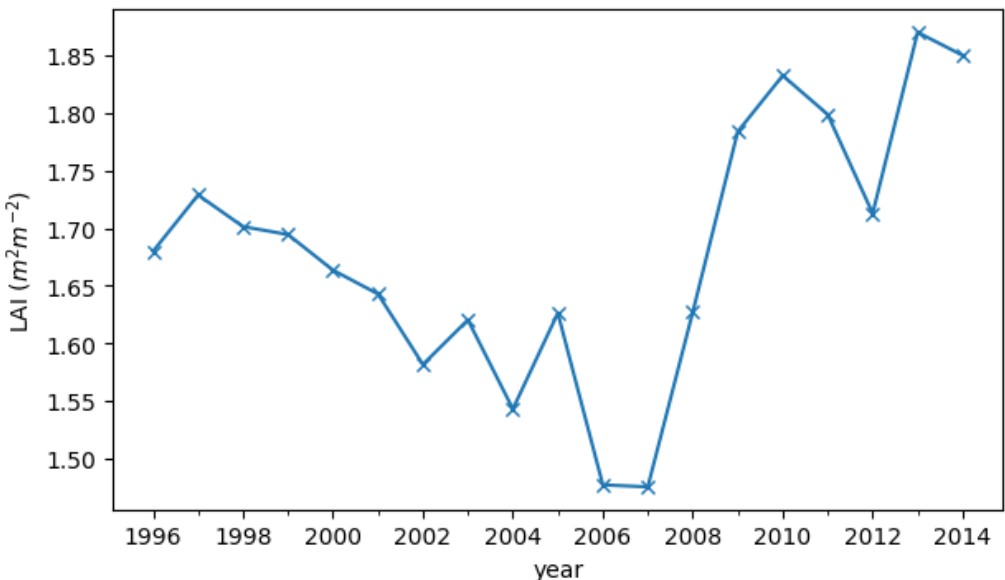

*Figure 6 Interannual variation in Leaf Area Index.*

*Table 7 Area, dry weight and concentrations of carbon, nitrogen and phosphorus in foliar samples from 1998 to 2024.*

| Year | Number of needles | Area | Dry weight | Leaf mass per area | C | N | P | C/N |
|------|------|------|------|------|------|------|------|------|
| | - | mm²/needle | mg | g/m² | g/kg | g/kg | g/kg | - |
| | | | | | | | | |
| 1998 | 39 | 140.3 | 31.5 | 220.0 | NA | NA | NA | NA |
| 1999 | 100 | 157.8 | 34.3 | 217.4 | NA | 19.2 | 1.0 | NA |
| 2000 | 100 | 123.0 | 26.7 | 217.4 | NA | 17.1 | 1.3 | NA |
| 2003 | 116 | NA | 16.9 | NA | 510.8 | 16.9 | NA | 30.4 |
| 2012 | 32 | NA | NA | 190.8 | 495.5 | 19.7 | 1.6 | 25.2 |
| 2024 | 300 | 125.9 | 26.0 | 206.5 | 528.8 | 17.8 | 1.4 | 29.7 |

### 3.1 Mean diurnal storage $CO_2$ flux

To understand how large the storage $CO_2$ flux is as a fraction of total $CO_2$ fluxes, the mean diurnal

variations of NEE (total flux), EC and storage measurement were calculated separately (Eq. (1)). Fig. 7

shows the magnitude of NEE (total flux) and the EC and storage components. The $CO_2$ storage flux is

typically in the order of 1 µmol m$^{-2}$ s$^{-1}$, but in extreme situations it can be between -5 and +3 µmol m$^{-2}$ s$^{-1}$.

At moments around sunrise and sunset, the EC fluxes are close to zero, while the (negative) storage flux

is largest around sunrise, indicating a release of carbon dioxides stored below the canopy at night.



Surprisingly, the release continues well until noon. In cumulative fluxes on a daily or longer timescale, the storage flux is negligible, but on hourly timescales the storage flux needs to be taken into account to represent the true ecosystem CO2 fluxes. Concludingly, the $CO_2$ storage flux is a significant but small fraction of the total NEE.

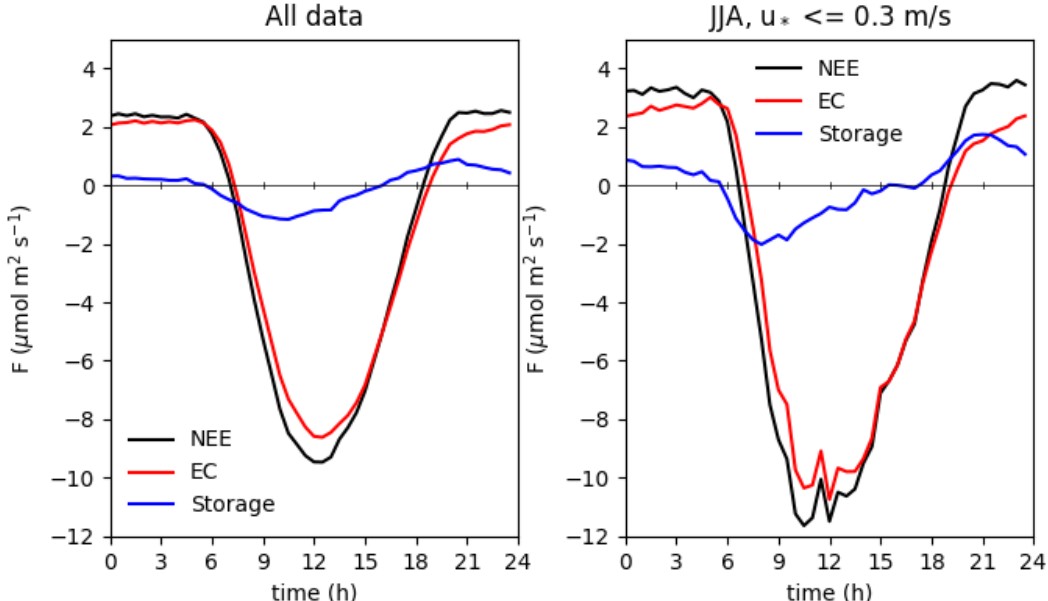

*Figure 7 Magnitude of NEE (total $CO_2$ flux), and the individual EC and storage components at a height of 27 meters, visualised as the mean diurnal cycle of all available data (left) and under conditions of low turbulence ($u^* < 0.3\ m\ s^{-1}$) and large respiration fluxes in summer (JJA denotes June, July and August)*

### 3.3 Seasonal and interannual variations in NEE

In an effort to verify the consistency of the EC dataset of carbon dioxide exchange over the years (Zhao et
al., 2025), we show the mean monthly NEE per year (Fig 8.), Subsequently, we calculated the monthly mean diurnal cycle, which we integrated to a monthly $CO_2$ exchange. Because we integrate the mean diurnal cycle, the storage component may be ignored and hence this estimate represents the monthly NEE. Here we implicitly assume an absence of lateral outflow of nocturnal storage fluxes. The figure shows a clear and consistent seasonal cycle, with carbon uptake from March to September and release
from October to February. The intensity of the winter respiration appears to decrease over time, whereas the intensity of the summer uptake is increasing. The mean annual uptake between 1997-2006 is around 350 gC $m^{-2}$ $yr^{-1}$ and between 2007 and 2016 around 550 gC $m^{-2}$ $yr^{-1}$. In 2018 and 2019 the mean uptake grew to an average of 820 gC $m^{-2}$ $yr^{-1}$, partially because of the reduction in wintertime net fluxes.

We stress that this dataset is based on eddy covariance measurements and is not gap-filled. Fig. 9 shows

the completeness of the dataset. Without $u^*$ filtering all months have a completeness larger than 85% of

the half hours. The $u^*$ filtering reduces the completeness to an average of 82%, not considering the

periods with fully lacking data. Upgrading Level 1 data into a Level 2 gap-filled dataset and analysing the

trend of NEE, total LE and H are beyond the scope of this study.

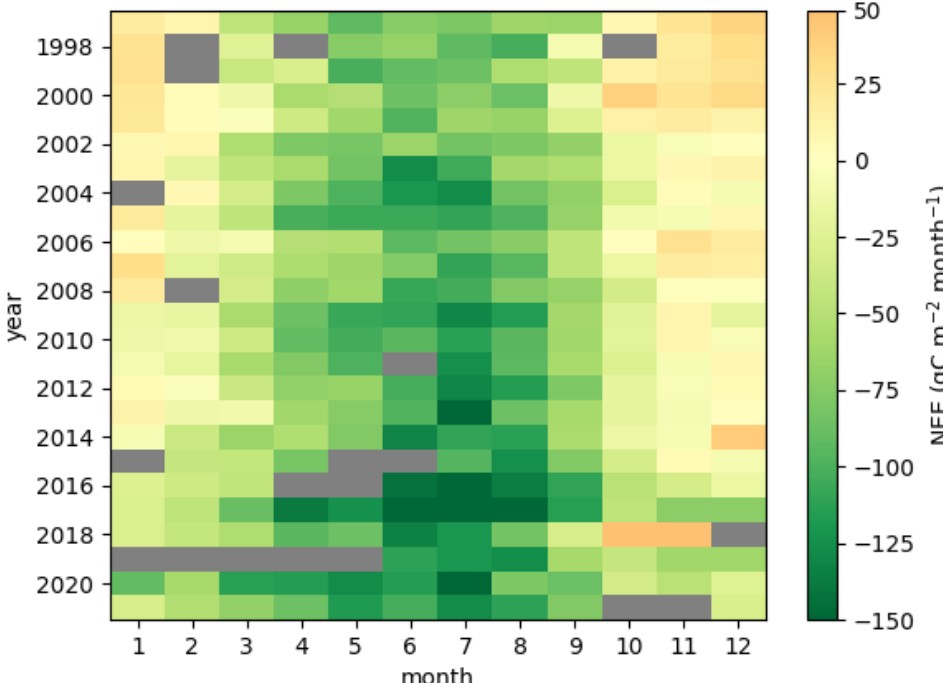

*Figure 8 Seasonal and interannual variation in net carbon dioxide exchange, based on the eddy covariance measurements. The values represent the monthly mean diurnal cycle integrated to a monthly total. Gray colors indicate a data availability less than 30%.*





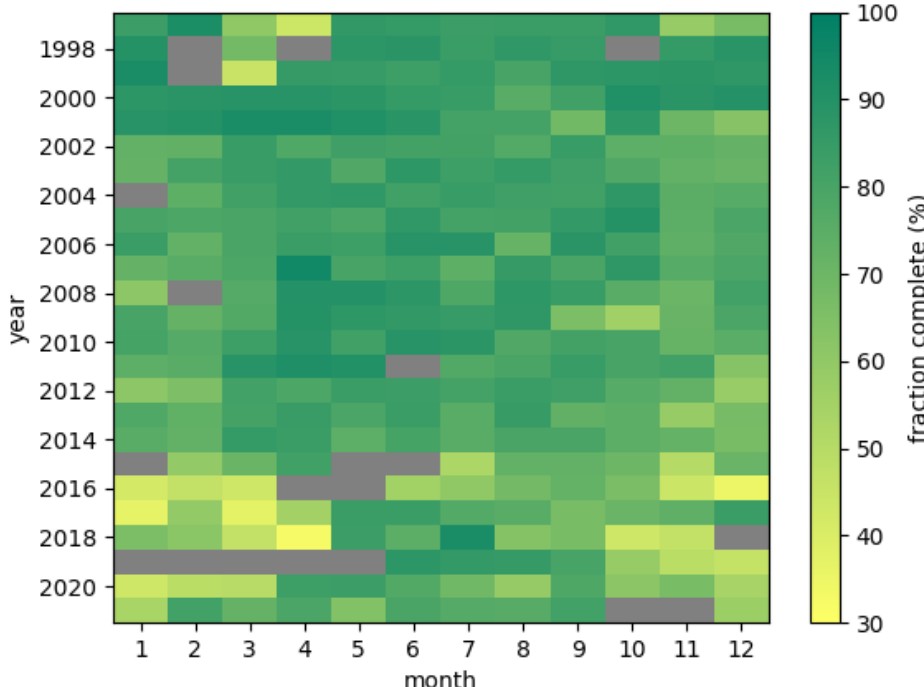

*Figure 9 Completeness of the eddy covariance carbon dioxide data. Gray colors indicate a data availability less than 30%.*


### 3.4 Energy balance residual

The energy balance closure ($R_n$-G vs. LE-H) was calculated. With the inclusion of storage fluxes of water and heat, Fig. 10 demonstrates an improved energy balance closure, supported by an enhanced regression relationship reflected in the slope and coefficient of determination ($R^2$) between available energy (i.e.,

Rnet-G) and turbulent fluxes (i.e., LE+H), which aligns with findings from previous studies (Leuning et al., 2012). The point colors indicate the hour of the day, suggesting that energy balance closure is lower in the mornings than in the afternoon, which could be related to unaccounted for energy storage in biomass. Detailed analysis of surface energy balance is beyond the scope of this study.



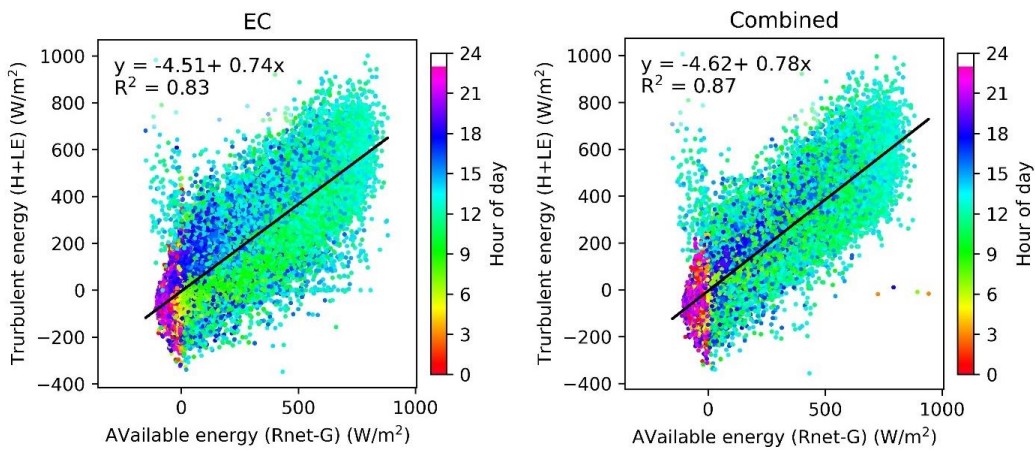

*Figure 10 Energy balance closure of the Loobos first tower datasets at a height of 27 meters. The left graph shows turbulent fluxes based solely on EC fluxes, and the right graph indicates turbulent fluxes combined with storage fluxes.*

## 4    Discussion and conclusion

Being one of the longest datasets of its kind (https://fluxnet.org/data/la-thuile-dataset/lathuile-data-

summary/, last access: 15 December 2024), this long and complete dataset can be used for further data

analysis, development and/or verification of models and validating satellite data retrievals. It is noted that

in 2021 a second tower was built and equipped next to the first. This second tower was labelled as an

Integrated Carbon Observation System (ICOS) class 2 Ecosystem site in 2023 (https://meta.icos-

cp.eu/resources/stations/ES_NL-Loo, last access: 15 December 2024) (Van Der Molen et al., 2025). The

data from the second tower may be regarded as a continuation of the dataset reported in this work and can

be accessed via the ICOS carbon portal (https://www.icos-cp.eu/, last access: 15 December 2024).

The Loobos tower has been running almost without major interruption for more than 25 years now,

serving as a robust platform for various scientific studies and educational activities. Over the years,

diverse experiments with different objectives have been conducted at and around the tower, ranging from

nitrogen deposition research (https://ruisdael-observatory.nl/wp-content/uploads/2018/12/Ewout-Melman-

PDF.pdf, last access: 15 December 2024), remote sensing studies (https://ruisdael-observatory.nl/wp-

content/uploads/2023/03/New-Loobos-ecosystem-site-compressed.pdf, last access: 15 December 2024) to

educational activities for field training courses at Wageningen University. In summary, in addition to its

role as a unique and well-equipped platform, the Loobos site offers a rich data set for continued research

and analyses.





As a component of the Ruisdael Observatory (https://ruisdael-observatory.nl/loobos/, https://maq-observations.nl/, last access: 15 December 2024), the Loobos Pine forest site represents one of the major land surface types in the Netherlands. It is typical for the extensive Veluwe region, which is aerodynamically rough, forested, located on well-drained sandy soils and vulnerable to summer drought

(Granier et al., 2007). The site is downwind from an area with intensive livestock farming and is exposed to high ammonia deposition. In combination with high $NO_x$ emissions from the cities and highways further upwind, high ozone and particulate matter (PM) model fractions may develop. Both the high input of reactive nitrogen, and the high ozone and PM concentrations may affect ecosystem growth (De Vries and Du, 2024; Visser et al., 2021; Visser et al., 2022; Grantz et al., 2003). To better understand these

dynamics, we intermittently measure ammonia dry deposition fluxes in cooperation with the National Institute for the Environment for Public Health and the Environment (RIVM) (Van Der Molen et al., 2025) and seek further funding to measure fluxes of reactive compounds.

Within the Ruisdael Observatory, the Loobos site, along with the Veekampen site (representing rural grassland (https://maq-observations.nl/, last access: 15 December 2024)) and the Cabauw site

(representing rural, grass and peat land (Bosveld et al., 2020)) collectively form a triangle network of comprehensive observation sites. This network provides valuable opportunities to understand changes in the land-atmosphere interaction and validate high resolution climate and land surface models.

### Data availability

The Loobos first tower dataset in Level 1 and ancillary data can be accessed at

https://doi.org/10.5281/zenodo.15721310 (Zhao et al., 2025) under a CC-BY4 open use license. The Level 1 data will also be available at the European Fluxes Database Cluster. The Level 2 gap filling and partitioning data based on the ONEFlux processing pipeline (Pastorello et al., 2020) for the FLUXNET release will be accessible at the ICOS carbon portal (https://www.icos-cp.eu/observations/carbon-portal) at the end of 2025.

### 530 Code availability

The Python codes for processing Level 0 into Level 1 dataset, and for plotting figures shown in the text can be found at https://git.wur.nl/zhao133/nl-loo_first_tower_project.git.

### Author contribution

JE, HD, EM and WJ built the first tower, collected data and maintained the site until 2018. JE, HD and

EM, BK, WJ and RH wrote the research proposal to secure funding for building-up and maintaining the tower. MvdM, BK and HS started maintaining the site and collected data in 2018. MvdM and HZ





conceptualized this study. HZ wrote the manuscript and verified the datasets. WP, MK and JV wrote the research proposal to get the Ruisdael funding for supporting the continuation of the Loobos infrastructure. All authors reviewed and edited the manuscript.

**Competing interests**

The contact author has declared that none of the authors has any competing interests.

**Acknowledgments**

We acknowledge the original AIRCOA design by NOAA (Britt Stevens) and adapted it for use with the CR100 logger by Jan Elbers. We thank University of Groningen for providing cylinders with calibration gases used in the AIRCOA system. We acknowledge the support of Wim Snijders in the field.

The Loobos measurements described in this paper were made possible through the grants below:

- The 'CarboEurope-Integrated project' supported by the European Commission through contract GOCE-CT2003-505572.
- The 'EUROFLUX project (ENV-CT95-0078)' funded by the European Union Fourth Framework Programme.
- The 'Infrastructure for Measurements of the European Carbon Cycle (IMECC) project' funded by the European Union (Framework Program 6).
- The 'GHG-Europe project' funded by the European Union (Framework Program 7).
- The 'Hydrology and water balance of forest in the Netherlands project' funded by the Dutch Ministry of Agriculture, Fisheries and Nature Management, the Dutch Forestry Commission (SBB).
- The 'Integrated observations and modelling of greenhouse gas budgets at the ecosystem level in the Netherlands project (ME1)' supported by the Dutch National Research Program Climate Changes Spatial Planning.
- The 'Climate Research Program on Climate Change of Wageningen University and Research' supported by the Ministry of Agriculture, Nature and Food Safety of the Netherlands.
- The Ruisdael Observatory funded by the Dutch Research Council (NWO) through a National Roadmap for Large-Scale Research Facilities.

## Appendix

### 0. Photos taken at Loobos site

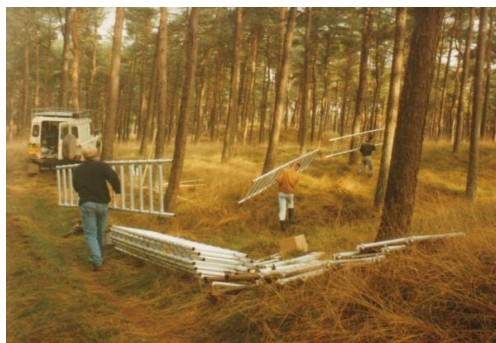 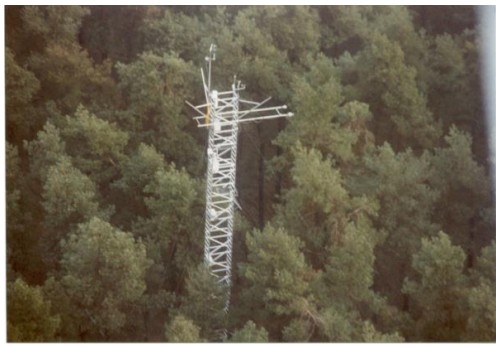

Figure A1-1. The left photo was taken in 1995 and the right photo in 2017.

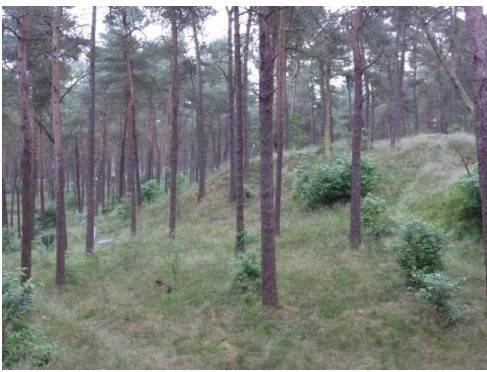 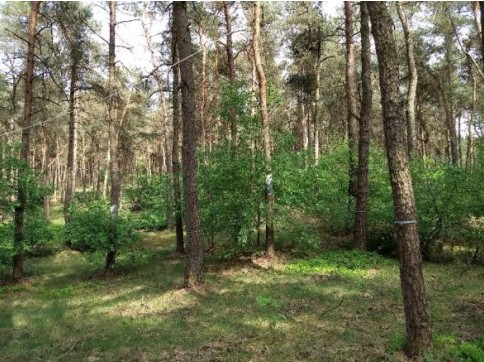

Figure A1-2. The left photo was taken in 2012 and the right photo in 2018.

Figure A1 Photos taken at the Loobos site.

### A. Profile CO₂ mole fraction measurements

The $CO_2$ profile measurements were calibrated twice a day (section 2.2.1). The table below shows the $CO_2$ mole fractions in the cylinders.

Table A1. $CO_2$ mole fraction of calibration cylinders provided by University of Groningen in the AIRCOA system deployed at the first tower in Loobos.

| Start date | H2 | H1 | L1 | L2 | LT |
|---|---|---|---|---|---|
| 2007-Jan-01 | 429.77 | 410.88 | 370.65 | 349.46 | 390.55 |
| 2012-Apr-11 | 537.14 | 430.57 | 375.46 | 328.83 | 390.55 |
| 2017-Mar-29 | 498.12 | 447.41 | 402.6 | 352.07 | 390.55 |
| 2020-Oct-27 | 464.62 | 418.5 | 387.91 | 367.93 | 390.55 |





**B. Water vapor pressure calculations**

The Maghus-Tetens empirical formula is used to calculate the pressure of saturated water vapor in air ($e_{sat}$) at a given temperature (T).

$$e_{sat} = e_o \times 10^{\frac{a \cdot T}{b+T}}$$  (A1)

$e_o$ is valued of 6.107 mbar for liquid water. $a$ and $b$ are constants specific to either water or ice. $a$ is valued of 7.5°C and $b$ of 237.3°C for water, and $a$ of 9.5°C and $b$ of 265.5°C for ice.

$$e_{actual} = \frac{RH}{100} * e_{sat}$$  (A2)

**C. AIRCOA CO₂ mole fraction calibration results**

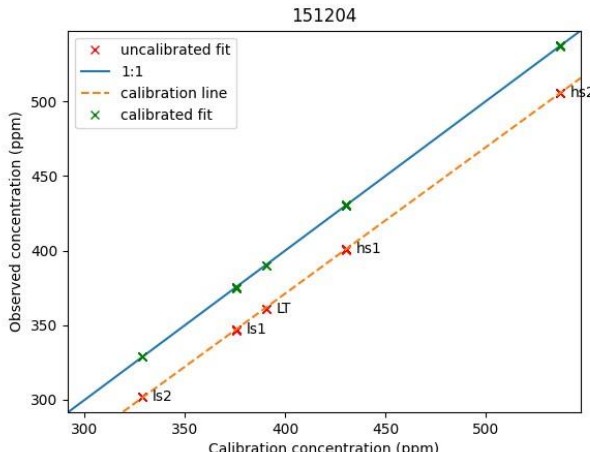

Figure A2. The calibrated CO₂ mole fraction against the observations on 12/04/2015 shown as an example. The ls and hs represent L and H calibration described in the text. The results indicate a good



calibration.

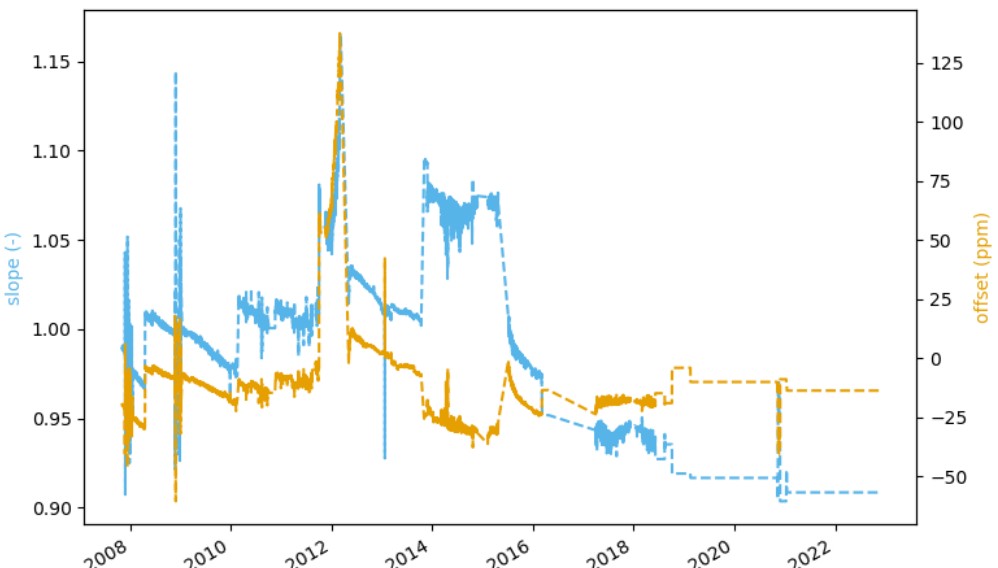


Figure A3. The calibration slope and offset in time series. It shows the value of the slope greater than 0.90 and smost values of the offset within 30 ppm.

**D. Calculations of CO₂ and H₂O storages under the canopy**

The integrated heat, CO₂ and H₂O model fraction under the canopy above the ground are calculated by equations below,

$$H\_content = \sum_{i=1}^{Z} \Delta z_i T_i \rho_i C_p \quad (i = 1, 2, 3, 4, 5, 6) \tag{A3}$$

$$LE\_content = \sum_{i=1}^{Z} \Delta z_i [H_2O]_i \rho_{air} C_p \gamma_{(T_i,p)} \quad (i1, 2, 3, 4, 5, 6)$$

$$CO_2\_content = \sum_{i=1}^{Z} \Delta z_i \frac{[CO_2]_i}{M} \rho_{air} \quad (i = 1, 2, 3, 4, 5, 6)$$

$$CO_2\_content\_2.5m = \sum_{i=5}^{Z} \Delta z_6 \frac{[CO_2]_i}{M} \rho_{air} \quad (i = 5, 6)$$





where $[H_2O]_i$ and $[CO_2]_i$ mole fraction measurements were conducted at five layers of 25.97 ($i = 1$), 7.5, 5.0, 2.5, 0.4 ($i = 5$) m, $i = 1$ denotes the ground layer, and $T_i$ were measured at the three levels (23.5, 7.5 and 5.0 m). The integration starts from the ground level of 0 m. $\rho_{air}$ denotes air density, which is determined by air temperature, water vapor pressure and air pressure. $C_p$ denotes the specific heat of air (at a constant pressure of 1005.0 mbar) J (kg$^{-1}$ K$^{-1}$). $\gamma_{(T_i,p)}$ denotes a rate of change related to the latent heat of vaporization, which is determined by air temperature $T_i$ and air pressure $p$, $M$ denotes the molecular weight of dry air at 0.028966 kg mol$^{-1}$. $H\_content$ and $LE\_content$ have a unit of J/m$^2$, and $CO_2\_content$ and $CO_2\_content\_2.5m$ have a unit of μmol m$^{-2}$.

The integrated heat, CO₂ and H₂O fluxes under the canopy above the ground are calculated by equations below,

$$SH\_1\_1\_1 = H\_strg = \frac{H\_content_{t+1} - H\_content_t}{1800} \qquad (A4)$$

$$SLE\_1\_1\_1 = LH\_strg = \frac{LH\_content_{t+1} - LH\_content_t}{1800}$$

$$SC\_1\_1\_1 = CO_2\_strg$$
$$= \frac{[CO_2\_content]_{t+1} - [CO_2\_content]_t}{1800}$$

$$CO_2\_strg\_2.5m$$
$$= \frac{CO_2\_content\_2.5m_{t+1} - CO_2\_content\_2.5m_t}{1800}$$

Where $t$ refers to time and $t + 1$ denotes next time stamp. $H\_strg$ and $LH\_strg$ have a unit of W m$^{-2}$, and $CO_2\_strg$ and $CO_2\_strg\_2.5m$ have a unit of μmol m$^{-2}$ s$^{-1}$.


**E. Physical ranges for assuring data quality**

Table A2 Values of physical ranges for assuring the quality of data from NL-Loo_BM and NL-Loo_Profile  streams. Information about variable names can be found in the description Excel sheet.

| Variable | Unit | Data_stream | Max | Min | Sdmax | Difmax | Difmin |
|---|---|---|---|---|---|---|---|
| SW_IN_1_1_1 | W m$^{-2}$ | NL-Loo_BM | 1000 | 0 | 400 | 500 | 0 |
| SW_OUT_1_1_1 | W m$^{-2}$ | NL-Loo_BM | 200 | 0 | 50 | 50 | 0 |
| LW_IN_1_1_1 | W m$^{-2}$ | NL-Loo_BM | 450 | 180 | 40 | 100 | 0.0001 |
| LW_OUT_1_1_1 | W m$^{-2}$ | NL-Loo_BM | 500 | 180 | 10 | 50 | 0.0001 |
| G_1_1_1 | W m$^{-2}$ | NL-Loo_BM | 50 | -10 | 2 | 4 | 0 |





| | | | | | | | |
|---|---|---|---|---|---|---|---|
| fapp | W m⁻² | NL-Loo_BM | 50 | -10 | 2 | 4 | 0 |
| G_3_1_1 | W m⁻² | NL-Loo_BM | 50 | -10 | 2 | 4 | 0 |
| G_4_1_1 | W m⁻² | NL-Loo_BM | 100 | -20 | 2 | 6 | 0 |
| RH_1_1_1 | % | NL-Loo_BM | 100 | 20 | 10 | 25 | 0 |
| T_1_1_1 | °C | NL-Loo_BM | 35 | -20 | 2 | 5 | 0 |
| LW_T_BODY_1_1_1 | °C | NL-Loo_BM | 40 | -20 | 2 | 5 | 0 |
| LW_T_BODY_2_1_1 | °C | NL-Loo_BM | 40 | -20 | 2 | 5 | 0 |
| WS_1_1_1 | m s⁻¹ | NL-Loo_BM | 10 | 0 | | 5 | 0 |
| WD_1_1_1 | ° | NL-Loo_BM | 360 | 0 | | 360 | 0.0001 |
| P_1_1_1 | mm/30minute | NL-Loo_BM | 8 | 0 | | 8 | |
| PA_1_1_1 | hPa | NL-Loo_BM | 1040 | 900 | | 10 | 0 |
| PPFD_DIR_1_1_1 | µmol m⁻² s⁻¹ | NL-Loo_BM | 2000 | 0 | 800 | 1000 | 0 |
| PPFD_OUT_1_1_1 | µmol m⁻² s⁻¹ | NL-Loo_BM | 200 | 0 | 100 | 200 | 0 |
| PPFD_DIF_1_1_1 | µmol m⁻² s⁻¹ | NL-Loo_BM | 2000 | 0 | 800 | 2000 | 0 |
| H₂O_1_1_1 | µmol m⁻² s⁻¹ | NL-Loo_BM | 1250 | 100 | 100 | 250 | 0 |
| CO₂_1_1_1 | µmol m⁻² s⁻¹ | NL-Loo_BM | 30 | 11 | 5 | 5 | 0 |
| H₂O_1_1_1 | mbar | NL-Loo_Profile | 25 | 1 | | 5 | 0.001 |
| CO₂_1_1_1 | ppm | NL-Loo_Profile | 550 | 320 | | 100 | 0.1 |
| RH_ | % | NL-Loo_Profile | 100 | 0.1 | 25 | 0 | |
| WS_1_2_1 | m s⁻¹ | NL-Loo_Profile | 20 | 0.01 | | | |
| WS_1_3_1 | m s⁻¹ | NL-Loo_Profile | 20 | 0.01 | | | |
| TA | °C | NL-Loo_Profile | 40 | -20 | 5 | 0 | |
| LE_content | J.m-2 | NL-Loo_ST | | 1 | | | |
| CO₂_content | umol.m-2 | NL-Loo_ST | | 1 | | | |
| CO₂_content_2.5m | umol.m-2 | NL-Loo_ST | | 1 | | | |
| SC_1_1_1 | µmol m⁻² s⁻¹ | NL-Loo_ST | 3 | -5 | | | |
| H_1_1_1 | W m⁻² | NL-Loo_EC | 600 | -250 | | | |
| LE_1_1_1 | W m⁻² | NL-Loo_EC | 600 | -250 | | | |
| FC_1_1_1 | µmol m⁻² s⁻¹ | NL-Loo_EC | 20 | -40 | | | |
| H₂O | mmol/mol | NL-Loo_EC | 25 | 1 | 5 | 0.001 | |
| CO₂ | ppm | NL-Loo_EC | 600 | 320 | 100 | 0.1 | |
| T_SONIC_1_1_1 | °C | NL-Loo_EC | 50 | -20 | 5 | 0 | |
| WS_1_1_1 | m s⁻¹ | NL-Loo_EC | 10 | 0.01 | | | |
| WD_1_1_1 | ° | NL-Loo_EC | 360 | 0 | 360 | | |
| ZL_1_1_1 | - | NL-Loo_EC | 10 | -10 | | | |
| USTAR_1_1_1 | m s⁻¹ | NL-Loo_EC | 3 | 0 | 1 | | |
| TS | °C | NL-Loo_Soil | 40 | -20 | 5 | 0 | |
| SWC | m³ m⁻³ | NL-Loo_Soil | 1 | 0 | 0.5 | | |





**F. Mean diurnal variations**

$$MDV = \frac{1}{N_t} \sum_{i=1}^{N_t} NEE_{t,i}$$
(A5)

Where $NEE_{t,i}$ refers to $NEE$ value at time $t$ on day $i$. $N_t$ denotes the numbers of days with $NEE$ data at time $t$.



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
