# Peer review of "The Loobos ecosystem first tower dataset: meteorology, turbulent fluxes and net ecosystem exchange (1996 to 2021)"

_Earth System Science Data, 2025_

## Author Comment (AC1)

**1st Reviewer's comments**

Dear Reviewer,

We sincerely appreciate the time and effort you have devoted to reviewing our paper. Your constructive comments have been helpful in improving the quality of the paper. We have revised the manuscript accordingly and provided our detailed responses to your comments below.

Best regards

Hong, on behalf of all co-authors.

**General comments.**

The manuscript "The Loobos ecosystem first tower dataset: meteorology, turbulent fluxes and net ecosystem exchange (1996 to 2021)" by Zhao et al. describes an impressive bulk of data collected over 25 years at one of the earliest micrometeorological towers installed in Europe for the systematic monitoring of CO2 and energy fluxes. The dataset includes, beyond flux variables, numerous parallel measurements of biophysical parameters and biological processes of the forest ecosystem such as LAI, foliage chemistry, biomass carbon stocks, sap flow density and water table depth.

A dedicated section explains in detail the flux data processing pipeline and the quality check procedures for flux and partly for meteorological variables. Finally, a synthesis of the dataset is illustrated by plots showing the consistency of the flux data, the mean diurnal variation and in particular the mean monthly NEE fluxes from which the trend towards a stronger carbon sequestration can be observed.

The presented dataset is undoubtedly of interest to the readers of ESSD and the well-structured manuscript delivers an overall clear and detailed description of it. Said that, I recommend the publication of the manuscript but only after a minor revision mainly to: (i) fix some issues concerning the terminology used to describe the ecosystem-atmosphere $CO_2$ exchanges; (ii) provide more information about the quality of meteorological measurements and the instruments' maintenance routines.

**Specific comments.**

Introduction section (ca. L60): I suggest including a map of the area showing the location of the measurements (eddy covariance, meteorological, forest transects, sap flow, etc.).

Reply: We thank the reviewer for this helpful suggestion. We have added a map into the text showing the locations of various measurements conducted at the first tower.

[Figure]

Figure 1 A map of measurement locations at the Loobos first tower. The first tower is in the center (0,0) m. The black square delineates the area where the tree inventories were done in 2000, 2005, 2008, 2012 and 2025. The green dots show the locations of the trees in the inventory.

Introduction – par 2.1.1: Some instruments were operated continuously for long periods of time (eg. pyranometers, quantum sensors, etc.). Report about the maintenance routine and its frequency to ensure accurate measurements, mentioning if the sensors were periodically calibrated and if the calibration drift was accounted for in processing the data.

Reply: We agree on the importance of sensor maintenance and calibrations. Indeed,  we did sensor calibration for key instruments such as the gas analyzer sensors, and we also conducted the routine maintenance on a bi-weekly basis, including visual inspection, cleaning of sensor domes and diffusers of remote dust and other deposits and verification of sensor leveling and alignment. These manual interventions ensured optical and structural integrity, while the long-term consistency of the data was further supported by the high intrinsic stability of the sensors themselves.

**We have added the descriptions below to the last paragraph of 2.1.1.**

"Notably, the key instruments that operate continuously over long periods, such as pyranometers, Vaisala instruments and quantum sensors were maintained on a bi-weekly scale to ensure high-quality and reliable data. Routine maintenance includes visual inspection, cleaning of sensor domes and diffusers of remote dust and other deposits and verification of sensor leveling and alignment.

While these manual interventions ensured optical and structural integrity, the long-term consistency of the data was further supported by the high intrinsic stability of the sensors themselves. For instance, air temperature measurements through Vaisala remained highly stable due to inherent resistance to calibration drift from the platinum resistance thermometers (PT100). Similarly, soil moisture and temperature sensors (Campbell Scientific, Inc.) described in section 2.1.4 demonstrated high temporal stability with negligible sensor drift throughout the study period. While the Vaisala humidity sensors drifted, unfortunately, they were not often calibrated.

The sonic anemometers and wind vanes were characterized by high long-term stability, as their measurement accuracy is dependent on fixed transducer geometry rather than electronic calibration. Any potential measurement deviations were attributed to frame-induced flow distortions rather than transducer degradation; therefore, the anemometer did not require factory calibration.

Radiation sensors were subject to slow, systematic sensitivity degradation due to the environmental aging of the sensor's optical surfaces. To account for this, these instruments were returned for factory recalibration every couple of years. To assess the potential for long-term sensitivity degradation in the radiation measurements, an intercomparison was conducted during an overlapping period (from 2022-Jan-21 to 2023-May-29). Incoming shortwave radiation at the first tower was compared against a secondary, independent sensor (Kipp & Zonen) deployed at an adjacent site ((Van Der Molen et al., 2025), https://meta.icos-cp.eu/resources/stations/ES_NL-Loo). The comparison result demonstrates high instrumental consistency, with relative differences remaining within 5% when averaged over a weekly time window (Figure S1). As such, the calibration drift was not applied in the data processing. ”

[Figure]

Figure S1 Relative difference of shortwave radiation measured from the first tower and the second tower.

**We have added the descriptions below to the last paragraph of 2.1.2.**

"Beyond these hardware transitions, the long-term continuity of the flux estimate was maintained through sensor-specific maintenance and calibration protocols designed to mitigate instrument drift.

For LI-COR infra-red gas analyser sensors, the closed-path LI-6262 was subjected to zero drift; this offset primarily affected absolute concentration values rather than EC flux calculations from high-frequency fluctuations. In contrast, the open-path LI-7500 series demonstrated high gain stability attributed to their robust optical design. Diurnal drift was assumed to be minimal and was further accounted for through standard coordinate rotation and detrending procedures during post-processing. These instruments were calibrated on an annual basis, supplemented by regular cleaning of optical windows to prevent signal attenuation from environmental debris. "

L223: I guess that the "depth of 15 cm" refers to insertion of the collar into the soil and not to the depth of the chamber which would be placed on top of the collar. Please confirm.

Reply: Yes, that's right. We revised it into "At each point, the collar was inserted into the soil with a depth of 15 cm and the soil respiration chamber was then placed onto the collar. "

L444:  the negative storage at sunrise should not be associated only to a "release of carbon dioxide" to the atmosphere, but also to the photosynthetic uptake of $CO_2$ by the vegetation below the eddy covariance sensors height.

Reply: We have revised it into "while the negative storage flux is largest around sunrise, indicating a release of carbon dioxides stored below the canopy at night. At the same time, plant photosynthetic uptake of CO2 begins, partially offsetting this upward flux. "

L460: fig. 8 shows mean monthly NEE, that is, by definition, the net flux resulting from the difference between the processes of photosynthesis and respiration. Therefore, in the case of this evergreen needle forest, I would not talk about a decrease of "winter respiration" but rather of the intensity of the $CO_2$ source.

Reply: We have revised "winter respiration" into "the $CO_2$ source".

Appendix (L570-571): Replace "left photo" and " right photo" with "photo on the right" and "photo on the left" respectively.

Reply: Done.

---

## Author Comment (AC2)

**2nd Reviewer's comments**

Dear Reviewer,

We sincerely appreciate the time and effort you have devoted to reviewing our paper. Your constructive comments have been helpful in improving the quality of the paper. We have revised the manuscript accordingly and provided our detailed responses to your comments below.

Best regards

Hong, on behalf of all co-authors.

Zhao et al gives a thorough description of the data collection and processing for data collected at the first Loobos tower. Data users will find this to be a comprehensive presentation of the sensor characteristics, location, and calibration.

It would be helpful to present the information about instrument changeover in a more easily digested way. For the eddy covariance measurements this is noted in the narrative text and Table 1, but showing this graphically would be easier for readers to absorb. Could you also comment on whether there was any overlap when sensor types were changed so that the data from each sensor could be compared. If there was no overlap, then note that in the text.

Reply: Thanks for your suggestions. We have added a figure showing an overview of the changeover information for the main instruments described in this study.

[Figure]

Figure A2 An overview of the changeover information for the main instruments deployed at the Loobos first tower site.

Unfortunately, there was no overlap when sensor types were changed. We added the sentence below to the end of section 2.1.2.

"Unfortunately, there were no overlapping measurements available during the anemometer replacement, precluding direct intercomparison between the two sensors. An overview of main instrument changes is presented in Fig. A2."

It is helpful to see some graphical examples of the typical diel patterns of CO2 flux. Could you include some examples of a mean $CO_2$ concentration profile, which would help assess whether the selected sampling heights were suitable for representing the profile shape and calculating the column integral?

Reply: We thank the reviewer for this suggestion. We have added a figure showing the mean diurnal cycles of the $CO_2$ mole fraction gradients ($dCO_2/dz$) as below. The figure demonstrates a coherent and physically consistent vertical structure, with the strongest gradient occurring near the surface during nighttime and early morning hours, and a progressively weaker gradient towards higher levels. In extreme situations, i.e. in the summer months June, July and August, when the respiration fluxes are large and with $u^* < 0.3$ m s$^{-1}$, the gradients are typically 1 ppm m$^{-1}$ between 25 and 7.5 m, 2 ppm m$^{-1}$ between 7.5 and 5.0 m and 5 ppm m$^{-1}$ below there. These findings indicate that the selected sampling heights adequately resolve the dominant features of the vertical $CO_2$ concentration profile relevant for calculating the column-integrated storage term.

We have added the above content to section 3.1.

[Figure]

Figure S2. Mean diurnal cycles of the $CO_2$ mole fraction gradients. Left: for all data with $u^* < 0.3$ m s$^{-1}$. Right: for all data in June, July and August with $u^* < 0.3$ m s$^{-1}$. The solid line refers to mean values and the dashed line denotes the mean $\pm$ 1 time standard deviation.

Other than that I just note a few typographical errors

Some instances at line 152 and thereafter where mole should replace model

line 444 carbon dioxide should be singular

Use of subscript in $CO_2$ should be consistent

Reply: Thank you. Done.